# Radiomics/Radiogenomics in Lung Cancer: Basic Principles and Initial Clinical Results

**DOI:** 10.3390/cancers14071657

**Published:** 2022-03-25

**Authors:** Athanasios K. Anagnostopoulos, Anastasios Gaitanis, Ioannis Gkiozos, Emmanouil I. Athanasiadis, Sofia N. Chatziioannou, Konstantinos N. Syrigos, Dimitris Thanos, Achilles N. Chatziioannou, Nikolaos Papanikolaou

**Affiliations:** 1Division of Biotechnology, Center of Systems Biology, Biomedical Research Foundation of the Academy of Athens (BRFAA), 11525 Athens, Greece; 2Clinical and Translational Research, Center of Experimental Surgery, Biomedical Research Foundation of the Academy of Athens (BRFAA), 11527 Athens, Greece; agaitanis@bioacademy.gr; 3Third Department of Internal Medicine, “Sotiria” Hospital, National and Kapodistrian University of Athens, 11527 Athens, Greece; yiannisgk@hotmail.com (I.G.); ksyrigos@med.uoa.gr (K.N.S.); 4Greek Genome Centre, Biomedical Research Foundation of the Academy of Athens (BRFAA), 11527 Athens, Greece; mathan@bioacademy.gr (E.I.A.); thanos@bioacademy.gr (D.T.); 5Nuclear Medicine Division, Biomedical Research Foundation of the Academy of Athens (BRFAA), 11527 Athens, Greece; sofiac@med.uoa.gr; 6First Department of Radiology, Medical School, National and Kapodistrian University of Athens, 11527 Athens, Greece; achatzi@med.uoa.gr; 7Computational Clinical Imaging Group, Centre for the Unknown, Champalimaud Foundation, 1400-038 Lisbon, Portugal; nickolas.papanikolaou@research.fchampalimaud.org; 8Machine Learning Group, The Royal Marsden, London SM2 5MG, UK; 9The Institute of Cancer Research, London SM2 5MG, UK; 10Karolinska Institutet, 14186 Stockholm, Sweden; 11Institute of Computer Science, FORTH, 70013 Heraklion, Greece

**Keywords:** radiogenomics, lung cancer, radiomics, learning algorithms, image science, review

## Abstract

**Simple Summary:**

Radiogenomics is a promising new approach in cancer assessment, providing an evaluation of the molecular basis of imaging phenotypes after establishing associations between radiological features and molecular features at the genomic–transcriptomic–proteomic level. This review focuses on describing key aspects of radiogenomics while discussing limitations of translatability to the clinic and possible solutions to these challenges, providing the clinician with an up-to-date handbook on how to use this new tool.

**Abstract:**

Lung cancer is the leading cause of cancer-related deaths worldwide, and elucidation of its complicated pathobiology has been traditionally targeted by studies incorporating genomic as well other high-throughput approaches. Recently, a collection of methods used for cancer imaging, supplemented by quantitative aspects leading towards imaging biomarker assessment termed “radiomics”, has introduced a novel dimension in cancer research. Integration of genomics and radiomics approaches, where identifying the biological basis of imaging phenotypes is feasible due to the establishment of associations between molecular features at the genomic–transcriptomic–proteomic level and radiological features, has recently emerged termed radiogenomics. This review article aims to briefly describe the main aspects of radiogenomics, while discussing its basic limitations related to lung cancer clinical applications for clinicians, researchers and patients.

## 1. Introduction

Cancer is a heterogeneous disease mainly caused by the accumulation of genetic and epigenetic changes, leading to aberrant cell proliferation. One of the fundamental aspects of understanding cancer pathobiology is correlating genomic alterations to cancer phenotypes. In this regard, the advent of DNA and RNA landscaping, further amplified by the human genome project and Cancer Genome Atlas, has directly linked genomic data to a vast array of tumour types [1,2]. Cancer imaging, which has traditionally been the cornerstone of disease diagnosis, staging, radiation planning and monitoring, has been lately supplemented by quantitative aspects, leading to image biomarker assessment called “radiomics” [3]. Combining the above modalities with underlying molecular features at the genomic–transcriptomic–proteomic level is known as “radiogenomics”, a novel approach that identifies the biological basis of imaging phenotypes [4].

Given the high availability of imaging data in patients with lung cancer, this specific cancer type has been “targeted” since the early days of radiomics and radiogenomics studies. In the current review, we will briefly describe the basic aspects of radiomics and radiogenomics and focus on clinical applications attracting considerable attention for reasons that we will extensively refer to in the following sections. In addition, we will discuss the challenges and problems of radiomics regarding their limited translation to the clinic and propose potential solutions to overcome these problems and therefore create added value for the management of lung cancer patients.

## 2. Radiomics Pipeline

With the development of texture analysis techniques, medical images are transformed into minable data by applying mathematical transformations on selected areas of tissues. A typical radiomics pipeline involves many steps [5]. Initially, a domain expert, such as a medical oncologist, needs to define the clinical use case, in other words, to formulate what the clinicians are expecting from the model to infer. Then, the relevant data source must be identified, and a bioimaging specialist will ensure that raw data are adequate for further analysis. Depending on data size, there are two options; in the case where the number of patients is a few dozens to a few hundred, the radiomics with classical machine learning (ML) algorithms should be explored first. However, if there are thousands of patients to be examined, deep neural networks could be the first choice. Following model training, validation should be made with data similar to the training cohort, the so-called internal validation, and independent data originating from other hospitals, the so-called external validation. In the following paragraphs, each pipeline step will be briefly discussed.

### 2.1. Defining the Problem at the Clinical Level

A successful ML application is primarily dependent on a solid, clinically meaningful formulation of the clinical use case. It is of paramount importance when considering different use cases to weigh the final decision not only on their clinical soundness, but instead on a realistic consideration of an adequate amount of data required for efficient training of the machine learning algorithm to produce a robust model that will be accurate and general [6]. The latter is certainly true when considering input data such as the medical images that are dynamic and change often based on information representation and lack of universally adopted standardised acquisition protocols [7]. In addition, the ground truth that will be selected must be robust with low variability, easily obtained, and not dependent on human interpretation. If the latter is inevitable, panels of experts must provide the ground truth to address interobserver variability effects.

### 2.2. Identification of Proper Data Sources

Although radiomics promotes the idea of a data-driven approach, specific domain expertise must be considered to select the meaningful potential data sources that will be recruited for further analysis. When imaging is considered, there is a preference for simplistic instead of “exotic” types of images such as conventional T1, T2, FLAIR and post-contrast images in magnetic resonance imaging (MRI) instead of an advanced biomarker imaging, such as for example, microvascular fraction quantified from an intravoxel incoherent motion diffusion experiment. Standard-of-care imaging approaches are also preferable since they are the imaging milestones in routine clinical protocols and therefore can guarantee the availability of high patient numbers.

### 2.3. Data Acquisition and Standardisation

All modern imaging modalities are employed in radiomics. Computed tomography (CT), MRI, positron emission tomography (PET) and single photon emission computed tomography (SPECT) are some of the clinical modalities widely employed for textural features extraction. The variety of scanners allow for a wide range of acquisition and image reconstruction protocols, and thus, the standardisation of all these protocols is necessary [8]. Furthermore, defining the accepted range of values for any textural features remains an open issue. The requirement behind data standardisation is to reduce changes due to technical details that introduce numerical variations to textural features as they are falsely associated with biological effects. For this reason, significant efforts have been undertaken towards the development of guidelines related to the acquisition protocols, the segmentation of tissues and the computation of radiomic features [9,10].

#### Tumour Segmentation

Following image acquisition, segmentation is one of the most critical and challenging phases for calculating textural features. There exist many challenges related to segmentation, including the increased inter-reader variability and the fact that it is a time-consuming process; even in the presence of automated artificial intelligence (AI)-powered algorithms, human intervention to correct and validate the outcomes is still necessary. Due to the latter, it is recommended that two or more readers should perform segmentation to smooth out subjective opinions regarding the exact borders and remove features that are sensitive to slight differences in the definition of the segmentation masks (Figure 1).

### 2.4. Feature Computation

There are three main types of radiomics features related to first level, shape and texture [11]. First-level features provide information about signal intensities across the region or the volume of interest. They are similar to the histogram metrics when signal intensity is represented in the histogram. They include mean, median, standard deviation, min, max, skewness, kurtosis, etc. Shape-related features provide structural information of the lesion and inform for two- and three-dimensional morphological characteristics. A larger surface-area-to-volume ratio, for example, is indicative of a more spiculated tumour, which is known to have more malignant potential in comparison to a round mass with a smaller ratio [12]. Thus, shape features can differentiate malignant lung nodules from benign ones. Besides, volume estimation has been shown to be better in evaluating treatment response than conventional methods [13,14,15,16]. Finally, texture- and gradient-related features refer to the quantitative ability to retain spatial relationships and interactions between pixel intensities in a given local neighbourhood [17]. Texture features are known to measure tumour heterogeneity and have been identified as the most closely correlated to outcomes in lung cancer.

### 2.5. Feature Selection

In radiomics, as in other omics applications, one of the most critical problems that needs to be addressed is called *p* >> *n*, which means that the number of patients available is far lower than the number of imaging features we can extract from each patient. The process that identifies the perfect balance between the number of features and the corresponding model performance is called feature selection, which is an essential step in machine learning. Various methods, including filtered, wrapper and embedded, are used to address the dimensionality problem, where in principle, they aim to pick up the most stable, informative features to be considered for model training. An essential part of developing a radiomic signature relates to the fidelity of radiomic features that should be assessed in various levels, including robustness, temporal and spatial stability, and reproducibility. For that purpose, dedicated phantoms can perform quality control of radiomic features and compare features from different centres, with different equipment.

Feature selection is achieved most of the time in different waves. A typical workflow in the first phase permits only stable features to be forwarded. A zero or near-zero variance method removes useless features. A correlation analysis removes redundant features, and finally, a more sophisticated method such as mRMR or RFE is used for slimming down our radiomic signature.

### 2.6. Model Selection/Assessment

According to ML best practices, several algorithms should be applied, including, among others, logistic regression, naïve Bayes, random forests, support vector machines, and boosted trees, to the training set and then the appropriate algorithm that performs best on the validation set should be selected. The latter can be used to make decisions related to the structure of the model, such as how many and which features will be used, the kinds of resampling and transformations that should be applied, and finally, the fine-tuning of the algorithm hyperparameters. We conclude with a single finalised model with an estimated prediction error following this process. A model should be applied to the independent dataset originating from different institutions to assess the generalisation error. The latter served as an indication of the model’s capability to perform equally well on external data (external validation) instead of the initial validation performed with data derived from the same institution, known as “internal” validation. It is essential to mention that the metric to be used to assess the model’s performance should be appropriate to the specific requirements of the use case. For example, accuracy as a performance metric can be misleading in a highly imbalanced problem. Depending on the cost of false positives and false negatives, someone should use f1, f0.5 or f2 scores.

## 3. Clinical Applications of Radiomics/Radiogenomics in Lung Cancer

In principle, all available modern digital imaging modalities can be used to perform radiogenomics modelling. Specifically, in the case of lung cancer, the most common modalities are CT and PET–CT, given the leading role they have in the clinical diagnostic workup of patients with lung cancer.

Zhou et al. [18] demonstrated an existing correlation between CT Hounsfield attenuation measurements and lesion margins with cell-cycle genes. They noted that the presence of irregular borders and ground-glass opacities in the lesion correlated with EGFR expression. In a study by Rizzo et al. [19], EGFR mutation was shown to be associated with CT features such as the presence of air bronchogram, pleural retraction, small lesion size, and absence of fibrosis. In contrast, *ALK* mutation was associated with pleural effusion. Round shape, nodules in non-tumour lobes, and smoking were variables linked to KRAS mutation. Other, less frequent mutations such as RET and ROS1, which comprise 1–2% of all lung adenocarcinomas, have also been assessed for associations with imaging features [20].

In a study of 26 patients, Gevaert et al. [21] discovered that the presence of air bronchograms was associated with overexpression of the K-ras oncogene. Weiss et al. [22] also found that imaging features could predict the K-ras status of patients with NSCLC. As a predictive marker, KRAS has been linked to round shape, nodules in non-tumour lobes, and multiple small nodules, as well as general radiomic profiles. Some studies reported predicting EGFR, but not KRAS. *ALK* rearrangement was linked to pleural effusion and lobulated margins [23,24,25].

By far, the best-characterised genetic mutation in NSCLC is EGFR (both exon 19 and exon 21 mutations) linked to contrast, Laws-Energy, median Hounsfield unit (HU), SUVmax, pleural retraction, small size, speculation, irregular nodules, poorly defined margins, ground glass, emphysema, locoregional infiltration, and “normalised inverse difference moment”, as well as combined radiomic profiles [26,27,28,29,30,31,32,33].

In the case of solitary pulmonary nodules, and their evaluation using 18F FDG PET, the recently published scientific results reveal that the shape and texture analysis obtained with radiomics analysis can lead to a better discrimination between benign and malignant lung nodules. More specifically, Palumbo et al. [34] examined PET–CT images from 111 patients and found that CT- and PET-based shape and texture features increase the accuracy of the prediction models. Furthermore, Caruso et al. [35] analysed CT-guided lung biopsy images of patients that had undergone a PET–CT scan with SUV less than 2.5, and they found that CT texture analysis can provide better discriminating information even in those cases without a significant variation to glucose metabolism. Finally, Zhou et al. [36] analysed PET and CT images from 769 patients, and they found that the obtained radiomic features, combined with machine learning methods, can be a discriminator tool between primary and metastatic lung lesions.

Radiogenomics aims to study the relationship between radiological and genomic features to noninvasively reveal potential biological features closely related to clinical outcomes. In a recent study, Li et al. subjected subsolid nodules from 154 lung adenocarcinoma patients to whole exome-sequencing while collecting all patient clinical and radiological data [37]. Their analyses determined that EGFR showed the most common significant mutation followed by aberrations in RBM10, TP53 and KRAS. The latter radiogenomics study revealed remarkable genomic heterogeneity in subsolid nodules, whilst inculpating CT data presented the opportunity to more accurately predict lung adenocarcinoma’s malignant degree and mutation characteristics.

In the relevant study of Zhou et al., the gene expression profiles of 113 non-small cell cancer patients were linked to their CT imaging features aiming to create a radiogenomic map [18]. Their approach revealed ten metagenes framing several molecular pathways with a predominant role of the EGFR pathway in lung cancer pathobiology by utilising RNA-sequencing technologies. Based on these features, a detailed radiogenomic map was created depicting correlations between imaging features and metagenes, showing that specific characteristics such as nodule attenuation and ground-glass opacity are associated with late-cycle genes and genes of the EGFR pathway, respectively.

Furthermore, Li et al. evaluated potential radiogenomics associations in solid lung adenocarcinoma by studying quantitative parameters obtained by dual-energy spectral CT combined with rearrangements in the three genes most involved in lung cancer molecular background [38]. A total of 96 patients were assessed for EGFR, KRAS and *ALK* mutations and rearrangements. These parameters were associated with quantitative parameters such as water and iodine concentration CT value at 70 keV. It was shown that spectral CT has the potential to predict KRAS and EGFR mutations as well as *ALK* reordering in solid lung adenocarcinoma, thus accurately reflecting molecular features of this tumour.

Gevaert et al. [21] succeeded in identifying prognostic imaging biomarkers in non–small cell lung cancer (NSCLC) by correlating clusters of co-expressed genes (metagenes) with PET–CT imaging features. This radiogenomics strategy was applied to a cohort of 26 patients with NSCLC, including (a) gene expressions, (b) a set of 180 computed tomography (CT) and positron emission tomography (PET) image features. They concluded that there exist 243 statistically significant pairwise correlations between image features and metagenes with an accuracy of 59–83%, while when the predicted image features were mapped to a public gene expression dataset with survival endpoints, several features including tumour size, edge shape, and sharpness ranked the highest for prognostic significance.

## 4. Definition of the Clinical Problem

Lung cancer is the most aggressive cancer regarding overall survival amongst all other cancer types globally. The choice of treatment depends on the stage of the NSCLC, the performance status, the comorbidities, the histological type and the molecular status, all of which are crucial factors determining the treatment choice.

Thoracic surgery is the recommended treatment for patients with stage I–II non-small-cell lung cancer (NSCLC), with a 5-year survival of 77–92% for clinical stage IA, 68% for stage IB, 60% for stage IIA, and 53% for clinical stage IIB. Even if an early stage of NSCLC appears simple to handle, physicians consider two issues when treating patients. First, there are patients with early-stage NSCLC that have a higher risk of recurrence based on tumour size, e.g., >4 cm, local spread, or other factors, e.g., pure solid type on high-resolution computed tomography (HRCT) [39], for which adjuvant therapy after surgery could lower the risk of recurrence. However, there is no direct correlation to the patient profile that could be benefited by adjuvant treatment. In addition, there are patients with early clinical stage NSCLC who have medical contraindications to surgical resection or who refuse surgery. For these patients, stereotactic body radiation therapy (SBRT) can lead to high local tumour control, >85% at five years, and low toxicity [40]. In this case, radiogenomics can offer additional means by providing individual information rather than population information of identifying and selecting patients best fit for adjuvant therapy or providing insights for prognosis of survival after treatment.

Grove et al. [41] found that measures of heterogeneity such as spiculation and entropy gradients are strong prognostic indicators of overall survival (OS) in patients with early-stage lung cancer. Aerts et al. [42] described a combination of features (size, shape, texture and wavelets) that could predict outcomes in patients with lung cancer. Studies from Huang et al. [43], Raghunath et al. [44], and Depeursinge et al. [45] found a correlation between their radiomic biomarker on CT and disease-free survival (DFS). Parmar et al. [46] found that their marker (comprising size, intensity, shape, texture and wavelet features) is associated with lung cancer prognosis, stage and histology, and Coroller et al. [47] provided a signature that correlated to distant metastasis.

## 5. Stage II NSCLC

During the last two decades, it has become clear that in early-stage II NSCLC adjuvant therapy applied after surgery is of benefit for patients with N1 and N2 disease (stage II and III), resulting overall in 4–5% absolute survival improvement at five years [38,48]. These results were obtained by administering cisplatin-based doublets, delivering at least three to four cycles. Newer adding agents, such as docetaxel, gemcitabine or pemetrexed, showed comparable efficacy.

NSCLC is only moderately sensitive to chemotherapy, with single-agent response rates in the range of 15% or better. Agents (e.g., gemcitabine, pemetrexed, docetaxel, vinorelbine) have shown promising single-agent activity, with response rates from 20–25% [49]. Patient selection criteria, such as proper recovery from surgery, the interval between surgery and the start of therapy, and the absence of significant comorbidities, are essential in awaiting the treatment benefit [48].

In this realm, radiomics is already making progress in predicting responses to adjuvant therapy. Two independent studies have suggested a potential role for radiomics in predicting pathological response to neoadjuvant chemotherapy before surgery, based on pre-treatment CT images of NSCLC [50,51]. It remains to be established whether radiogenomics can provide additional or complementary data to basic clinical decisions to select the proper patients for the appropriate adjuvant setting.

## 6. Locally Advanced-Stage III and Metastatic-Stage IV NSCLC

Treatment for stage III NSCLC requires a multidisciplinary approach involving a thoracic surgeon, a radiotherapist, a pulmonologist, a radiologist and an oncologist. Treatment options depend on the size of the tumour, the location of lesions, which lymph nodes it has spread to, the patient performance status, comorbidities, and how well it is foreseen to tolerate treatment. At this stage, treatment may help the patients live longer and feel better by relieving symptoms, but it is not likely to cure the disease. In most cases, relapse occurs within 1–2 years; thus, radio genomics could individualise the treatment approach by identifying certain imaging factors that could affect each patient’s response rate and overall survival depending on the genomic features of individual tumours. In our days, there is also the option for immunotherapy (a fully human, immunoglobulin G1 kappa (IgG1κ) monoclonal antibody that selectively blocks the interaction of PD-L1 with PD-1 and CD80). Immunotherapy (Durvalumab) is given for a year, as a consolidation therapy, after the completion of concurrent chemo-radiotherapy for unresectable stage III NSCLC. Thus, the knowledge of radiomics/radio genomics could improve the prognostic and predictive value of NSCLC stage III is crucial.

Unfortunately, most cases of NSCLC are diagnosed at stage IV. From this starting point, the hardships for the clinicians are many.

First, the primary and metastatic lesions coexist in the imaging appearance. There is extended heterogeneity within the lesion and among lesions in terms of imaging characteristics and mutation index. Not all lesions can be biopsied to identify the primary and secondary mutations.

Next, genotyping can be possible or not for insurance reimbursement issues. Even where possible, it will often be reimbursable for one gene/mutation. At the same time, it is recommended that screening for a specific set of mutations is what best identifies the patient to benefit most from targeted therapies [49].

Third, immunotherapy for lung cancer (mainly immune checkpoint inhibitors) may potentially achieve durable responses, however, only in a subset of patients. Still, the majority of NSLC patients do not respond to immunotherapy. Notably, the pattern of responses to immunotherapy varies, including in the standard RECIST criteria, pseudoprogression and hyperprogression. Reported rates of pseudoprogression are <10%, and rapid progressions, called hyperprogression, are reported to range from 4% to 29%. Some patients may present dissociated responses, with some lesions shrinking and others growing. In such cases, local treatment with surgery or radiotherapy for growing lesions may be considered [52]. Last, immunotherapy comes along with toxicity. A total of 73.4% of patients present any immunotherapy-related adverse event, and 26.6% present with a grade 3 or higher adverse event [53]. Monitoring for immunotherapy related pneumonitis, which when untreated can be life-threatening, is mandatory to safeguard NSCLC patients.

In this context, radiogenomics studies are anticipated to shed light and support monitoring patients, allocating them to the most suitable therapeutic option available and predicting response to treatment, the prognosis of survival, or monitoring disease control.

Computed tomography (CT) studies have provided evidence supporting the clinical decision. As mentioned above, the best-characterised genetic mutation in NSCLC was EGFR (both exon 19 and exon 21 mutations) linked to contrast [28], Laws-Energy [29], median Hounsfield unit (HU) [30], SUVmax [54], pleural retraction [14], small size [14,23], speculation [23], irregular nodules [18], poorly defined margins [18], ground glass [18,31], emphysema [32], locoregional infiltration [32], and “normalised inverse difference moment” [26], as well as combined radiomic profiles [24,25,33,55,56].

As a predictive marker, KRAS has been linked to a round shape [14], nodules in non-tumour lobes [14], and multiple small nodules [23], as well as general radiomic profiles [24,25]. ALK rearrangement was linked to pleural effusion [14] and lobulated margin [57]. Halpenny’s [58] study showed that *ALK*-positive tumours tended to be more prominent with more solid consistency and involved more thoracic lymphadenopathy. This study showed no correlation between *ALK* mutation and the presence of pleural effusion on CT. HER2, a gene often amplified during acquired resistance to EGFR-targeting therapy, was also studied [57].

Most studies derived radiomics signatures in radiotherapy planning or diagnostic images acquired before therapy. Nearly all studies evaluated patients undergoing treatment with cytotoxic chemo-radiotherapy. More recently, several studies have assessed the potential of radiomics to improve patient stratification for targeted therapies and immunotherapy agents [29]. For example, Tang and colleagues linked radiomic features to a tumour immune phenotype in patients with stage I–III NSCLC, finding patients with heterogeneous tumours, which correlated with low PD-L1 and high CD3 cell count, had better prognosis [59].

There are 24 CT studies evaluating how radiomic signatures of NSCLC relate to genomics [24,29,60,61,62,63,64,65,66], signalling pathways [67] and histopathology [67,68,69,70,71,72,73,74,75,76,77]. For example, Rios Velazquez and colleagues found distinct imaging phenotypes for EGFR and KRAS mutations from CT images of patients with NSCLC [24]. Some studies that relate radiomics to patient outcomes also relate their radiomic signature to genomics [45] or biological markers.

Collectively, these 64 studies present a positive and encouraging view of the potential for radiomics signatures to deliver personalised medicine. However, two important limitations are readily apparent. First, while nearly all studies report at least one positive association between CT radiomic signature and outcome (OS, PFS, recurrence or toxicity) or tumour biology (genomic or pathology biomarkers and signalling pathways), the particular radiomic signature derived from these analyses varies substantially between studies. Consequently, few study signatures are directly comparable with one another, and thus the literature does not identify specific candidate radiomic signatures suitable for further extensive multicentre evaluation.

In conclusion, from the radiologists and clinicians perspective, radiomics/radiogenomics could better determine the accuracy of malignancy of pulmonary nodules, which have been detected by CT scan in order to treat curatively, select patients with early-stage lung cancer who are appropriate for post-surgical treatment, and determine patients with stage III NSCLC who can tolerate immunotherapy as consolidation therapy after concurrent treatment with chemotherapy-radiation therapy. Moreover, radiomics/radiogenomics could support treatment selection by excluding NSCLC patients sensitive to hyperprogression in immunotherapy, thus avoiding an expensive and without survival benefit treatment option. They could also predict oncological outcomes such as response rate, progression-free survival and overall survival according to the baseline imaging appearance. All the above medical achievements declare that radio genomics could guide more personalized patient care.

## 7. Discussion

### 7.1. Challenges

Although radiomics appears to be straightforward and simple, many challenges should be addressed. Three essential concepts need to be carefully considered regarding the data: quantity, quality, and diversity. Radiomics is a data-hungry methodology, and in the presence of regulatory directives such as GDPR or HIPPA, easy unobstructed flow of data between institutions is difficult. High quality curated data has fundamental value for every radiomics project. The concept “garbage in garbage out” applies here; thus, we need to make an effort to gather, clean, transform and curate the data to be eligible for value creation. There are some considerations regarding the number of patients that should be used in these studies. The smaller the patient sample, the less robust the developed models; therefore, they quickly become overfitted. This is a phenomenon when the model explicitly learns the patterns found on the training data, failing to make accurate predictions on slightly different data than the training set used to build the model. In addition, the smaller the dataset used, the less representative it can be on the real-world data. It is vital to allow the ML algorithms to be exposed to diverse datasets to extract robust and reproducible patterns.

Today, there is a clear translation gap of radiomics to clinical practice due to the lack of adequate data and limited interpretability of current models. Thus, it is difficult to conceptualise how a model makes a specific prediction, compromising patient safety. In the first phase of radiomics (Radiomics 1.0), the focus was on maximising the model’s performance, often using models that are black boxes, such as neural networks. The latter harmed the capability of these models to be trusted by their end users since it was impossible to explain the inner mechanics of decision making of such models. However, several recent methods, known as explainable AI (xAI), have been developed and gradually explored and integrated into radiomics. SHAP and LIME are among the most popular methods that satisfy, to a certain extent, the high performance that black box models offer with both local and global explainability capabilities. SHAP is based on game theory, where each feature in the model has a different contribution influencing the final prediction. SHAP values reflect the latter concept and represent valuable tools to understand the decision-making process at a global level.

### 7.2. How to Move to Radiomics 2.0

Another common misconception of Radiomics 1.0 was that all efforts regarding model development were finalised as long as the training and validation of a model were completed. A growing number of investigators criticising the norm was based on a combination of utilising historical, retrospective data for training a model and a small finite independent cohort often called the “external” to test the model. People are shifting in the somewhat overlooked area of model deployment. Instead of training and testing the model with “outdated” retrospective data, “live” real-world data should also participate in both tasks through a continual or incremental learning scheme. Thus, using static instead of adaptive learning strategies in AI is somewhat contradictory. Quoting Stephen Hawking, “Intelligence is the ability to adapt to change”. It is well known that medical images have been designed to produce eye-pleasing pictures rather than to satisfy any basic rule that all quantitative methods respect. In addition, imaging modalities are constantly updated at a hardware and software level, causing an interesting effect that can be observed only after deploying the model. The latter is called “data drift”, and in principle, it can be responsible for the gradual deterioration of the model’s performance. There is a great need to monitor the model’s behaviour and possibly organise retraining sessions with contemporary data to keep the model in good “health” and deliver its promise. Another area that has also been significantly neglected is model usability. When designing a radiomics project, the first thing is to ask the end users and gather their input on how they need to interact with the model, obtain its predictions and exploit them in a meaningful way. Failure to identify all these concepts will significantly reduce the possibility of using the model; therefore, minimal clinical value will be generated.

## 8. Conclusions

Radiomics/radiogenomics could support treatment selection, excluding NSCLC patients sensitive to hyperprogression to immunotherapy, thus avoiding expensive and nonsurvivable benefit treatment options by predicting oncological outcomes such as response rate, progression-free survival and overall survival. All the above achievements declare that radiogenomics could guide more personalised patient care. However, there is still a long way until radiomics will be integrated into clinics and delivered to the patient, caregiver, or health authorities and deliver what they have promised. Inevitably, similar to all the other disruptive technologies, it must go through a maturation process. In our view, we are already there, prepared to face the challenges, since at least we may acknowledge them.

## Figures and Tables

**Figure 1 cancers-14-01657-f001:**
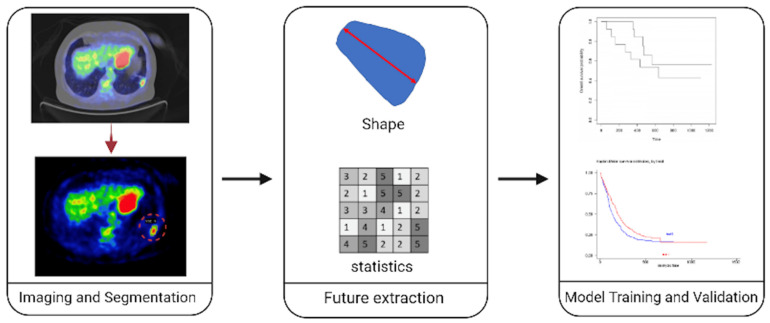
Typical workflow of a radiomic analysis. Initially, the proper medical imaging modality is selected (PET–CT in our example. Then, the radiologist segments the tissue of interest in all slices, resulting in a volume of interest. Consequently, radiomic features are computed using only the tissues included in the volume of interest. Such features can reflect shape information, signal intensities of tissues inside the VOI, and texture-related information that can reflect tissue heterogeneity. Finally, machine learning models are trained and validated to predict clinical outcomes or to classify patients according to genomic or molecular characteristics.

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
