# Peer review of "Radiomics/Radiogenomics in Lung Cancer: Basic Principles and Initial Clinical Results"

_cancers, 2022, doi:10.3390/cancers14071657_

Round 1

Reviewer 1 Report

The submitted paper seems to be original and well developed.

It has also a great scientific relevance since the topic is actual in scientific community.

Moreover, I retain that it is necessary to consider the increasing role of radiomics in nuclear medicine fields and for instance in the use of Radiomics in the evaluation of 18F FDG PET of solitary pulmonary nodules. According to recent literature, the shape and texture features obtained with radiomic from 18F-FDG PET/CT can lead to a better discrimination between benign and malignant lung nodules by increasing the accuracy of the prediction models by an appreciable margin.

Citing the follow references

1)Palumbo, B.; Bianconi, F.; Palumbo, I.; Fravolini, M.L.; Minestrini, M.; Nuvoli, S.; Stazza, M.L.; Rondini, M.; Spanu, A. Value of Shape and Texture Features from 18F-FDG PET/CT to Discriminate between Benign and Malignant Solitary Pulmonary Nodules: An Experimental Evaluation. Diagnostics 2020, 10, 696. https://doi.org/10.3390/diagnostics10090696.

2)Radiomics and functional imaging in lung cancer: the importance of radiological heterogeneity beyond FDG PET/CT and lung biopsy. Caruso D, Zerunian M, Daffina J, Polici M, Polidori T, Tipaldi MA, Ronconi E, Pucciarelli F, Lucertini E, Rossi M, Laghi A. Eur J Radiol. 2021 Sep;142:109874. doi: 10.1016/j.ejrad.2021.109874. Epub 2021 Jul 27.

3) Use of radiomics based on (18)F-FDG PET/CT and machine learning methods to aid clinical decision-making in the classification of solitary pulmonary lesions: an innovative approach. Zhou Y, Ma XL, Zhang T, Wang J, Zhang T, Tian R. Eur J Nucl Med Mol Imaging. 2021 Aug;48(9):2904-2913. doi: 10.1007/s00259-021-05220-7. Epub 2021)

Author Response

Response to Reviewer #1

We would like to thank the reviewer for this valuable comments. A relevant paragraph was added in the section "Clinical applications of radiomics/radiogenomics in lung cancer", page 5 (highlighted in yellow) including, as suggested, all the latest publications in this field.

Further three new References (35, 36, 37) (highlighted in yellow) were included in the Revised version of the manuscript.

Reviewer 2 Report

This review article aims to briefly describe the main aspects of radiogenomics while discussing its basic limitations related to lung cancer clinical applications for clinicians, researchers and patients. It has certain clinical guiding significance,however,there are still some deficiencies that need to be revised.

1.The logical framework is rather disordered, and the segmentation and classification of articles by clinical stage are too broad.

2.The clinical research part is mostly a list of data. It is suggested to add the author's comments for readers to think about;

3.There are many research data involved in the article. It is suggested that the data can be sorted into different categories to show the data of each research mentioned in the article, so as to make it more intuitive.

4.The language of the article needs further polishing.

Author Response

We would like to thank the Reviewer for his careful and constructive comments, that will help increase the impact of our article. According to the Reviewer’s comments we included the suggested changes in the revised version our manuscript and highlighted them in yellow in the text. Below please find a-point-by-point reply to your comments (shown in red fonts). We believe that our manuscript has certainly benefited from these suggestions.

Reviewer #2

This review article aims to briefly describe the main aspects of radiogenomics while discussing its basic limitations related to lung cancer clinical applications for clinicians, researchers and patients. It has certain clinical guiding significance,however, there are still some deficiencies that need to be revised.

1.The logical framework is rather disordered, and the segmentation and classification of articles by clinical stage are too broad.

2.The clinical research part is mostly a list of data. It is suggested to add the author's comments for readers to think about;

  1. We would like to thank you for this suggestion. In order to make the meaning clearer, we removed lines 20-21 (p.6). The removed text is the following:

“Radiomic features extracted from pre-treatment computed tomography (CT) images correlated with intensity-based fea-tures and NSLC patient survival (39)”.

  1. Same as above, we removed lines 24-34 (p.6):

“….whereas Ganeshan et al. (41) and Win et al. (42) found that textural features correlate with OS. Song et al. (43) describe wavelet features that correlate with OS. In contrast, Fried et al. (44) found textural features correlate to OS and locore-gional control and freedom from distant metastasis. Aerts et al. (45) described a combination of features (size, shape, tex-ture and wavelets) that could predict outcomes in patients with lung cancer. Studies from Huang et al. (46), Raghunath et al. (47), and Depeursinge et al. (48) found a correlation be-tween their radiomic biomarker on CT and disease-free sur-vival (DFS). Parmar et al. (39) found that their marker (com-prising of size, intensity, shape, texture and wavelet features) is associated with lung cancer prognosis, stage and histology, and Coroller et al. (49) provided a signature that correlated to distant metastasis.”

  1. Furthermore, we removed lines 46-53 (p.7):

“From 2015 to 2019, 41 CT studies were published that linked radiomics to lung cancer patient outcome. In general, these studies sought to evaluate whether or not radiomic sig-natures could outperform existing methods for patient risk stratification. 20 studies related radiomics to overall survival (30, 45, 63, 64, 65, 66, 67, 68, 69, 70, 71, 72, 73, 74, 75, 76, 77, 78, 79, 80), 18 to the likelihood of local or metastatic recur-rence (49, 67, 68, 69, 81,82, 83, 84 85, 86, 87, 88, 89, 90, 91, 92, 93), 6 to response, disease-free or progression-free survival (80, 52, 46, 94, 53, 95), and 2 to staging (96, 97). Two further studies focused on the association of radiomics signatures to lung toxicity (98, 99)”.

3.There are many research data involved in the article. It is suggested that the data can be sorted into different categories to show the data of each research mentioned in the article, so as to make it more intuitive.

According to the Reviewers’ suggestion, in page 8 (lines 17-22), the text was modified and the following paragraph was inserted:

“In conclusion, from the radiologists and clinicians perspective, radiomics-radiogenomics could determine better the accuracy of malignancy of pulmonary nodules, which have been detected by CT scan in order to treat curatively, select patients with early-stage lung cancer who are appropriate for post-surgical treatment, determine patients with stage III NSCLC who can tolerate immunotherapy as consolidation therapy after concurrent treatment with chemotherapy-radiation therapy. Moreover, radiomics/radiogenomics could support treatment selection by excluding NSCLC patients sensitive to hyper- progression in immunotherapy, thus avoiding an expensive and without survival benefit treatment option. They could also predict oncological outcomes such as response rate, progression-free survival and overall survival according to the baseline imaging appearance. All the above medical achievements declare that radio genomics could guide more personalized patient care”.

4.The language of the article needs further polishing.

Thank you for your valuable comment. The article was thoroughly revised by a native English speaker.

Reviewer 3 Report

The content of this manuscript is interesting, especially big data and machine learning are the upcoming trend for medical diagnosis and disease management. However, I hope the review could be much more informative. The association between radiological patterns (radiomics) and genetic profile of lung cancer (EGFR mutation, ALK mutation, etc) may not reflect the true pathobiology of cancers. I believe demographic factors (gender, age, and ethnicity) and treatment strategy could affect the radiomic of lung cancers. For example, when the authors relate immunotherapy side effects or clinical outcomes with radiogenomic, the authors should mention the types of immunotherapy (adoptive immune cell transfer, monoclonal antibodies, immune checkpoint inhibitors etc). Therefore, it would be nice for the authors to summarize the information in Clinical applications of radiomics/radiogenomics in lung cancer section to a table, to improve readability. Comments for minor amendment are:

  1. Figure 1 is too simple. Please briefly elaborate the workflow in figure legend.
  2. Please cite properly. For example, in page 7, change (30, 45, 63, 64, 65, 66, 67, 68, 69, 70, 71, 72, 73, 74, 75, 76, 77, 78, 79, 80) to (30, 45, 63-80).

Author Response

Reviewer #3

We would like to thank the Reviewer for his careful and constructive comments, that will help increase the impact of our article. According to the Reviewer’s comments we included the suggested changes in the revised version our manuscript and highlighted them in yellow in the text. Below please find a-point-by-point reply to your comments (shown in red fonts). We believe that our manuscript has certainly benefited from these suggestions.

The content of this manuscript is interesting, especially big data and machine learning are the upcoming trend for medical diagnosis and disease management. However, I hope the review could be much more informative. The association between radiological patterns (radiomics) and genetic profile of lung cancer (EGFR mutation, ALK mutation, etc) may not reflect the true pathobiology of cancers. I believe demographic factors (gender, age, and ethnicity) and treatment strategy could affect the radiomic of lung cancers. For example, when the authors relate immunotherapy side effects or clinical outcomes with radiogenomic, the authors should mention the types of immunotherapies (adoptive immune cell transfer, monoclonal antibodies, immune checkpoint inhibitors etc). Therefore, it would be nice for the authors to summarize the information in Clinical applications of radiomics/radiogenomics in lung cancer section to a table, to improve readability.

1.Regarding the types of immunotherapy, we added in page 7 (line 6) the additional information requested, starting from:

“…..genomic features of individual tumors….. . In our days there is also the option for immunotherapy (a fully human, immunoglobulin G1 kappa (IgG1κ) monoclo-nal antibody that selectively blocks the interaction of PD-L1 with PD-1 and CD80). Immunotherapy (Durvalumab) is given for a year, as a consolidation therapy, after the completion of concurrent chemo-radiotherapy for unresectable stage III NSCLC.So, the knowledge if radiomics/radiogenomics could improve the prognostic and predictive value of NSCLC satge III is crucial”.

  1. Regarding your suggestion form determining the type of immunotherapy in page 7, line 17 the highlighted text was added:

«..Third, immunotherapy for lung cancer (mainly immune checkpoint inhibitors) may potentially achieve durable responses, however, only in a subset of patients. Still, the majority of NSLC patients do not respond to immunotherapy..»

Comments for minor amendment:

Figure 1 is too simple. Please briefly elaborate the workflow in figure legend.

According to the Reviewer’s suggestion the Figure Legend was modified and the workflow was elaborated in greater detail.

Please cite properly. For example, in page 7, change (30, 45, 63, 64, 65, 66, 67, 68, 69, 70, 71, 72, 73, 74, 75, 76, 77, 78, 79, 80) to (30, 45, 63-80).

Citations throughout the text were corrected as suggested by the reviewer.

Round 2

Reviewer 1 Report

the manuscript  seems to be more suitable after the revision

Reviewer 2 Report

None.